# Testing of the Taxonomic Diagnosis of *Zebrus pallaoroi* Kovačić, Šanda & Vukić, 2021 (Actinopteri: Gobiiformes: Gobiidae), on a Large Sample from the Western Mediterranean

**Marcelo Kovačić** [1],*, **Lucas Bérenger** [2] and **Julien P. Renoult** [3]

1. Natural History Museum Rijeka, Lorenzov Prolaz 1, HR–51000 Rijeka, Croatia
2. BIOTOPE, 22 Boulevard Maréchal Foch, 34140 Mèze, France; lberenger@biotope.fr
3. CEFE, University of Montpellier, CNRS, EPHE, IRD, 34293 Montpellier, France; jurenoult@gmail.com
* Correspondence: marcelo@prirodoslovni.com; Tel.: +385-5155-3669

**Abstract:** A large sample of 65 individuals of the recently described goby *Zebrus pallaoroi* was collected in France. The species identity of these individuals was confirmed based on morphology. In addition, the species identity was validated through genetic analysis for one of the two specimens collected from the new depth records for the species. The diagnostic characters of *Z. pallaoroi*, originally based on a limited number of type specimens, were tested on this larger sample and critically analyzed. The diagnostic characters of *Z. pallaoroi* were revised, and recommendations were made for improving the study of diagnostic characters in gobiid species description, particularly when based on small sample sizes. The record of *Z. pallaoroi* in the western Mediterranean significantly expands its known geographic range, increases the maximum recorded depth of the species, and reveals syntopic co-occurrence with its phylogenetically closest relatives.

**Keywords:** *Zebrus pallaoroi*; new record; diagnosis; morphology; genetics; artificial habitats; France; Monaco

**Key Contribution:** The diagnostic characters of *Zebrus pallaoroi* were revised, its geographic range was significantly expanded, the species' maximum recorded depth was increased, and the species' syntopic co-occurrence with its phylogenetically closest relatives was revealed. Recommendations were made for improving the study of diagnostic characters in gobiid fish species description, particularly when based on small sample sizes.

## 1. Introduction

A taxonomic character is any attribute used to recognize, differentiate, or classify an animal taxon [1]. Taxonomic characters can be morphological (including coloration), physiological, molecular, behavioral, ecological, or geographic [2]. In principle, "the delimitation of a taxon is performed by the general description of a new species, i.e., by the complete statement of the characters of a taxon without special emphasis on those characters which distinguish it from coordinate units" [2]. However, species descriptions should also include a diagnosis [2]. The diagnosis provides the characters that differentiate the taxon from other taxa with which it might be confused [1]. Kottelat and Freyhof [3] even used the term "diagnosable" as a criterion for a species to be recognized. More precisely, the diagnosis should provide information on how the species differs from each of the most similar or closely related species for at least one unique, easily recognizable character state [4,5]. The diagnosis should also be concise, consisting of a minimal set of statements that offer clear and precise information [4,5]. The differential diagnosis is a formal statement of the characters that distinguish a given taxon from other specifically mentioned taxa [2]. The differential diagnosis is part of the diagnosis [2]. The differential characters against particular taxa can also be presented in detail separately after diagnosis, or even in the

remarks section of a species description, with the comparison of character values and states of new taxa against other specifically mentioned taxa, e.g., [6]. Unfortunately, the recommendation for a standardized diagnosis is still not part of the International Code of Zoological Nomenclature [1], although the commissioners of the International Commission on Zoological Nomenclature have advocated for a stricter definition of "species diagnosis" in future editions of the code [7]. As a result, taxonomy is filled with species descriptions that fail to provide contrastive comparisons with other species or to be state-specific in the character description, which frustrates later biologists' efforts to recognize species without having to consult the physical name-bearing type [7].

Since about 2000, taxonomists in general have increasingly tried to combine morphological and molecular data for detecting and delimiting species [8]. In zoology, the other types of characters (behavioral, ecological, or geographic) are traditionally rarely applied for species differentiation, but in recent times they have been increasingly used [9]. In current practice in gobiology, a new species is established as valid mostly based on morphological (including coloration) and molecular evidence that distinguishes it from closely related species (the examples of gobiid species description reviewed in [10]). Out of the 13 Mediterranean gobiid species described in the 21st century, only the earliest has a description including only morphological and coloration characters as evidence (reviewed in [10,11]). In contrast, the remaining twelve species were differentiated from closely related species using both genetic and morphological evidence (the gobiid species description references are available in [10] for the native species and [11] for the alien species described in the Mediterranean).

The first formal diagnoses of new animal taxa that included DNA-based characters came out in 2001 [8]. Anyway, some of the most recent fish species descriptions still rely only on morphology [12]. However, the prevailing practice when describing new European marine gobiid species is to list differential morphological and coloration characters in the species diagnosis, while molecular evidence is presented in a separate part of the species description (reviewed in [10,11]).

The morphological diagnosis usually contains the most useful information for the later identification of new and additional material [5]. Indeed, while the diagnosis is more complete and thus more reliable than identification keys [5], it is significantly shorter, less complex, faster, and easier to use than the complete morphological description or the molecular methods [4,5]. Diagnoses are highly advantageous, enabling simple and confident confirmation of identification after using a key [5]. Unfortunately, species are often described based on a limited number of type specimens and without morphological analyses of additional non-type individuals. Ideally, diagnostic characters should have states (for qualitative characters) and ranges (for quantitative characters) that do not overlap between the compared species [4]. However, even for the non-overlapping character states and ranges, there is no guarantee that the values measured in the studied individuals are representative of the species. For example, in all 13 gobiid species described since the beginning for the 21st century for the Mediterranean Sea, the external morphology was described from the type material only (reviewed in [10,11]). Nine species were described from six or less type specimens, and four species from eleven or more type specimens (the species description references are available in [10] for the native species and in [11] for the alien species described in the Mediterranean). For species descriptions based on a small number of individuals (e.g., fewer than six), it thus remains speculative whether the character states or ranges of the type specimens can be representative of the entire species. In addition, even a large sample, if collected from a single population of a geographically widespread species, may fail to capture the true variability of the species.

The gobiid genus *Zebrus* de Buen, 1930, was first erected as the subgenus *Zebrus*, including two species, later elevated to the rank of genus by [13]. One of the species, *Gobius thori* de Buen, 1928, was synonymized with *Thorogobius ephippiatus* (Lowe, 1839) [14]. Miller [15] then redescribed *Zebrus* as a monotypic genus including only *Zebrus zebrus* (Risso, 1827). Recently, Kovačić et al. [6] described a second species in this genus, *Zebrus*

*pallaoroi* Kovačić, Šanda, & Vukić, 2021, from the southern Adriatic, northern Ionian, and northern and western Aegean Seas. *Zebrus pallaoroi* and *Z. zebrus* differ in morphology and genetics. Kovačić et al. [16] presented additional records of this species, expanding its known distribution to the central and northern Adriatic Sea and increasing the amount of morphologically studied material of this species to seven specimens.

Despite the potential weaknesses of gobiid species diagnoses based on small samples, to our knowledge, no published examples exist re-evaluating or testing these diagnoses once larger samples have become available. Here, using the Mediterranean goby *Z. pallaoroi* as a case study, we aim to evaluate the robustness of a diagnosis established from a small number of type individuals when confronted with a larger set of new specimens (Figure 1). The diagnosis of *Z. pallaoroi* was originally based on three individuals, with a slight correction suggested by [16] following the analysis of four additional individuals. A large number of individuals of *Z. pallaoroi* were collected in France and Monaco by the authors (L.B. and J.P.R.).

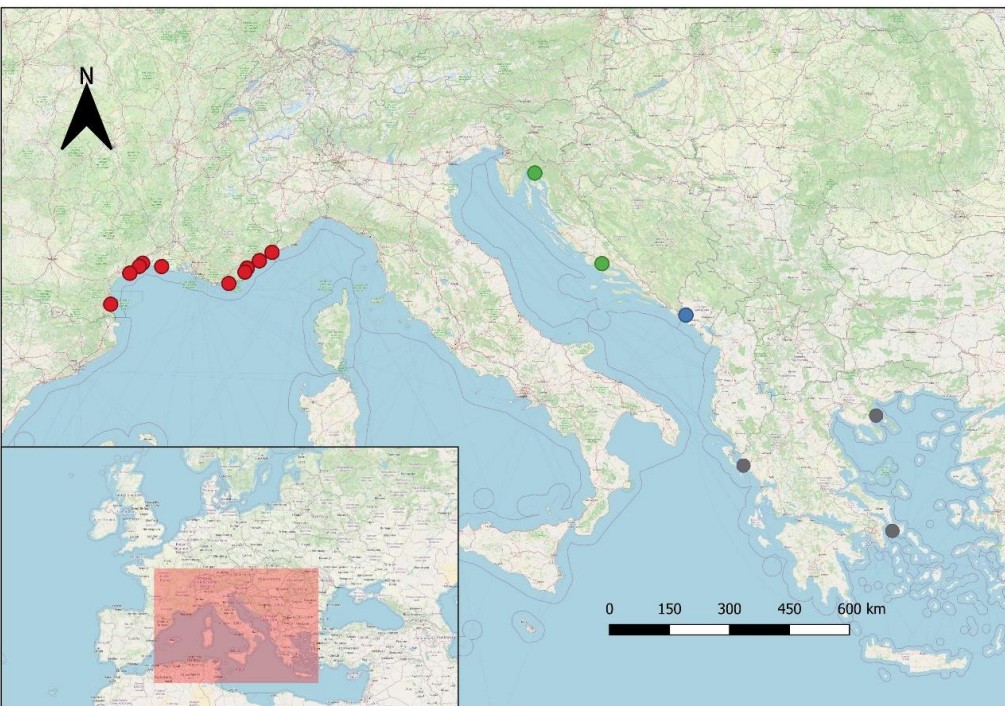

**Figure 1.** Known distribution of *Zebrus pallaoroi*. New records presented in this study (•), type locality (•), and other localities (•) from [6] and (•) from [16]. Map created using QGIS 3.22.3 software.

The objectives of this paper are to report the first record of *Z. pallaoroi* in the western Mediterranean, to expand knowledge of the species' geographical and ecological characteristics, and to test and revise its species diagnosis, originally based on the limited number of only diagnostic morphological characters, using a large sample from a population geographically distant from the type locality.

## 2. Materials and Methods

### 2.1. Sampling

We analyzed 97 individuals belonging to the genera *Millerigobius* Bath, 1973, or *Zebrus* based on the external appearance [6,16], collected between 2018 and 2022, along the Mediterranean coast of France and in Monaco. Specimens were collected from 12 localities, spanning from the Spanish border to the Italian border, and distributed across 22 samples. A locality corresponds to a geographic area spanning a few hundred meters (e.g., a harbor), while a sample is defined by the position of a meter-level precision and by the collection day. Of these 22 samples, 20 correspond to specimens collected during the maintenance of

Biohut© artificial habitats during 2018, 2021, and 2022, conducted under the supervision of LB. Biohut© habitats are designed for the restoration of coastal fish nursery functions in degraded environments. They are steel cages of approx. $1 \times 1 \times 1$ m in size, filled with oyster shells, and are typically suspended just below the water's surface. During maintenance, the Biohut© was removed from the water and all the shells were removed and cleaned. L.B. collected the cryptobenthic fish and donated its samples to J.P.R. in 2023. The remaining two samples were collected from natural habitats, with each sample containing one individual collected by J.P.R. with a hand net during recreational dives.

*2.2. Morphology*

In taxonomy, the term "character" is used in two distinct ways: either as the difference between taxa or as the structure that varies. In the first case, the term "signifier" describes the structure that varies and "character" is the difference between taxa [2,17]. Here, we follow the second approach, a "character" refers to the structure that varies, while a "character state" represents one of at least two specific variations, such as the blue or brown eye coloration [2]. A character state, as a qualitative character value, applies to categorial variables, whereas quantitative variables, whether discrete or continuous, are presented by with an individual value, which is, for samples or for populations, expressed as a range.

*Zebrus pallaoroi*, *Z. zebrus*, and *Millerigobius macrocephalus* (Kolombatović, 1891) were identified based on their morphological diagnoses [6,16,18]. We used 11 morphological characters actually used or just only considered for *Z. pallaoroi* morphological diagnosis [6,16] to identify *Z. pallaoroi*. These included eight characters from [6], though one of these characters, "the lateral dark brown bands on the body," was split into two separate characters: the number of dark lateral bands (photo 9 in Figure 2) and the mean width of the dark lateral bands compared to the mean width of the pale interspaces (photo 10 in Figure 2), resulting in nine characters. The tenth character is the body depth at the pelvic-fin origin, mentioned as non-overlapping between *Zebrus* species in the studied material in the original species description of *Z. pallaoroi* [6], but overlapping according to Miller's [15] values for *Z. zebrus* (photo 11 in Figure 2) and therefore not used in [6]. Finally, the eleventh character was the count of dark lateral bands in front of the second dorsal fin, which was proposed in [16] to replace the total count of dark lateral bands originally proposed in [6] (photo 8 in Figure 2). The 11 characters examined in this study are thus as follows: (1) eye diameter in snout length (SN/E); (2) anterior nostril length in posterior nostril length (PN/AN); (3) eye diameter in head length (HL/E); (4) presence of a short anterior transverse ridge that connects the left and right ventrolateral head ridges; (5) length of the ventral spine in anterior membrane depth at the midline (AM/VI); (6) diameter of the head canal pore $\alpha$ in the distance between pore $\rho$ and $\theta$ ($\rho\theta/\alpha$); (7) length of the suborbital sensory papillae row *c5i* compared to the distance between row *c5i* and row *d*; (8) width of the lateral dark bands at the lateral midline compared to the width of the pale interspaces at the lateral midline; (9) total count of dark lateral bands on the body counted at the lateral midline; (10) count of dark lateral bands in front of the vertical of the second dorsal fin, with the bar immediately behind the axilla counted but not the one that includes the second dorsal fin base; and (11) body depth at the pelvic-fin origin in standard length (SL/VD) (Figure 2). Morphometric measurements followed [6]. SN is measured from the anterior-most end of the upper lip to the anterior-most margin of the orbit including the ligamentous ring; E is the horizontal orbit diameter excluding the ligamentous ring around the eye; AN is measured at the nostril anterior edge from the bottom to the rim without including proliferations from the rim; PN is measured at the nostril anterior edge from the bottom to the rim; HL is measured from the anterior-most end of the upper lip to the posterior end of the opercular membrane; AM is measured from the bottom of the membrane to the edge at the midline; VI is measured from the bottom of the spine to its tip; $\alpha$ is the horizontal diameter of the head canal pore $\alpha$; $\rho\theta$ is the closest distance between pore $\rho$ and $\theta$; VD is measured at the level of the insertion of the pelvic spine; SL is measured from the median anterior point of the upper lip to the

base of the caudal fin (posterior end of the hypural plate). The exact position of landmarks used for measurements are illustrated in Figure 2. Measurements smaller than 20 mm were taken with interactively selected points in the Olympus cellSens Entry 2.2. software using the SC180 camera and U-TV0.5XC-3 camera adapter on the stereomicroscope SZX10 of the same producer, while the standard length, as the only measure out of this range, was taken using a digital caliper with a resolution of 0.01 mm viewed under stereomicroscope magnification. Except SL, all other pairs of the compared measures were taken using the same objective magnification to avoid potential bias, even though a calibration was performed for each magnification at the start of the measurements. The terminology of the lateral line system followed [10]. The specimen lengths are presented as the standard length (SL) + caudal-fin length (CL). The sex of the specimens was determined based on the shape of the urogenital papillae. The specimens were deposited at the Natural History Museum Rijeka.

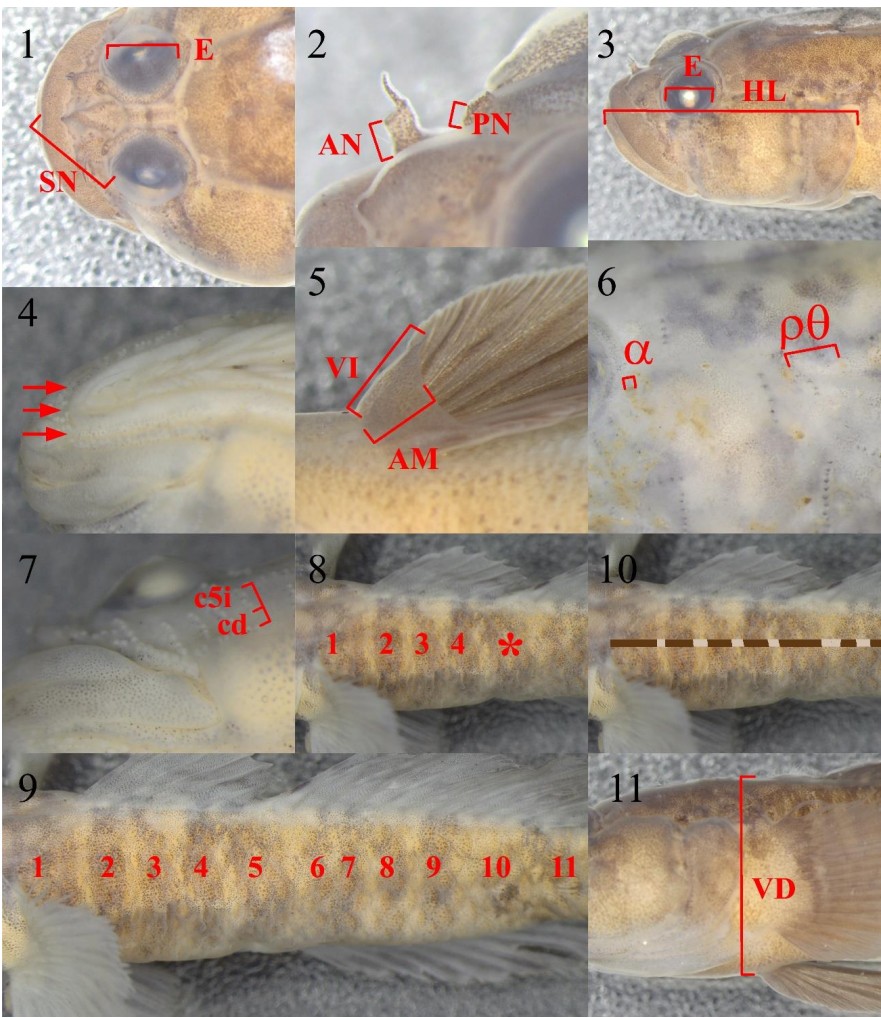

**Figure 2.** Illustration of the diagnostic characters investigated in this study: (**1**) snout length (SN) and eye diameter (E); (**2**) posterior nostril length (PN) and anterior nostril length (AN); (**3**) head length (HL) and eye diameter (E); (**4**) short transversal ridge connecting the left and right ventrolateral head ridges on the anterior part (marked with arrows); (**5**) anterior membrane in midline depth (AM) and the length of the ventral spine (VI); (**6**) head canal pore $\alpha$ diameter ($\alpha$) and distance between pore $\rho$ and $\theta$ ($\rho\theta$); (**7**) length of the suborbital sensory papillae row *c5i* (c5i) and the distance between row *c5i* and row *d* (cd); (**8**) lateral dark band widths compared to pale interspace widths; (**9**) total count of lateral dark bands on the body; (**10**) count of lateral dark bands in front of the vertical of the second dorsal fin, * anterior-most band below the second dorsal fin; (**11**) body depth at the pelvic-fin origin (VD).

Statistical analyses were performed using Python (version 3.11.8) with pandas (version 1.5.3; [19]) and scipy.stats (version 1.9.3; [20]) libraries for data processing and statistical analyses, respectively.

### 2.3. Molecular Analysis

We sequenced the cytochrome b mitochondrial gene for the individual PMR VP5841 (GenBank accession n°: PQ505129).

DNA was purified using the 96-Well Plate Animal Genomic DNA Miniprep Kit (Biobasic, Markham, ON, Canada) from a muscle sample taken before the specimen was fixed in 4% formaldehyde and stored in 96° ethanol. PCR amplification was performed in 5 μL of Multiplex PCR kit (Qiagen, Hilden, Germany) master mix using the primers AJG and H5 and the protocol detailed in [6]. PCR products (3 μL) were verified by electrophoresis in a 1.5% agarose gel. The PCR products were then paired-end sequenced using primers ZzebF1 and ZzebR1 [6] and an Illumina MiSeq system by Eurofins Genomics (https://www.eurofinsgenomics.eu/, accessed on 13 December 2024).

## 3. Results

### 3.1. Material

Among the 97 individuals examined, 65 were identified as *Z. pallaoroi* (Figure 3), 10 as *Z. zebrus,* and 20 as *M. macrocephalus* (Table 1; see justification in Morphology in the Section 3). One early juvenile and one overly damaged adult could not be identified. We determined 32 males and 33 females for *Z. pallaoroi*, 4 females and 6 males for *Z. zebrus*, and 7 females and 13 males for *M. macrocephalus*. The size of *Z. pallaoroi* ranged from 22.07 + 5.06 mm to 34.19 + 8.13 mm for females, while the standard length of the smallest male was 20.97, having a damaged caudal fin that could not be measured, and the largest male size was 44.06 + 11.41 mm (Table 2). The size of *Z. zebrus* ranged from 22.01 + 5.96 mm to 25.36 + 5.76 mm for females, and from 21.31 + 5.55 mm to 30.17 + 7.19 mm for males (Table 2). Finally, the size of *M. macrocephalus* ranged from 22.10 + 5.74 mm to 34.01 + 8.35 mm for females, and from 19.50 + 4.36 mm to 29.25 + 7.51 mm for males.

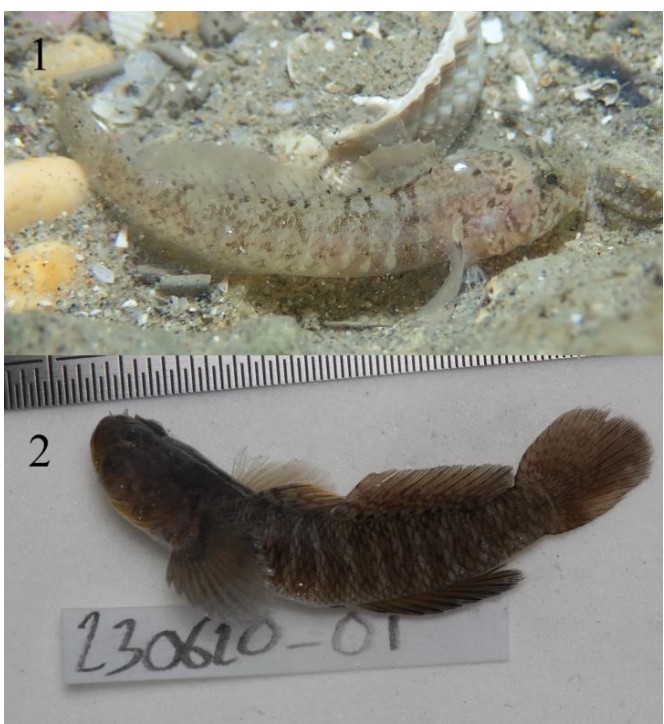

**Figure 3.** *Zebrus pallaoroi*, male, 33.08 + 8.54 mm, PMR VP5841 Prévost, Palavas-les-Flots, France. (**1**) Live specimen; (**2**) freshly collected specimen. Photos by J. Renoult.

**Table 1.** The collected samples of *Zebrus pallaoroi*, *Zebrus zebrus*, and *Millerigobius macrocephalus*.

| Catalog Number | Species | No. of Individuals | Collecting Point, Locality | Date | Latitude | Longitude |
|---|---|---|---|---|---|---|
| PMR VP5841 | *Z. pallaoroi* | 1 | Prévost, Palavas-les-Flots | 20 June 2020 | 43.515786 | 39. 05692 |
| PMR VP5842 | *Z. zebrus* | 2 | Fontvieille A, Monaco | 1 July 2021 | 43.729322 | 7.422575 |
| PMR VP5843 | *Z. pallaoroi* | 2 | Fontvieille A, Monaco | 1 July 2021 | 43.729322 | 7.422575 |
| PMR VP5844 | *Z. pallaoroi* | 10 | Camille Rayon Golf Juan A, Vallauris (Golfe-Juan) | 24 June 2021 | 43.562951 | 7.077255 |
| PMR VP5845 | *M. macrocephalus* | 6 | Camille Rayon Golf Juan C, Vallauris (Golfe-Juan) | 24 June 2021 | 43.567066 | 7.079608 |
| PMR VP5846 | *Z. pallaoroi* | 8 | Camille Rayon Golf Juan C, Vallauris (Golfe-Juan) | 24 June 2021 | 43.567066 | 7.079608 |
| PMR VP5847 | *Z. pallaoroi* | 3 | Fréjus D, Fréjus | 22 June 2021 | 43.421977 | 6.748286 |
| PMR VP5848 | *Z. pallaoroi* | 3 | Fréjus C, Fréjus | 22 June 2021 | 43.420939 | 6.747772 |
| PMR VP5849 | *M. macrocephalus* | 1 | Fréjus A, Fréjus | 22 June 2021 | 43.418976 | 6.74892 |
| PMR VP5850 | *Z. zebrus* | 1 | Fréjus A, Fréjus | 22 June 2021 | 43.418976 | 6.74892 |
| PMR VP5851 | *Z. pallaoroi* | 2 | Fréjus A, Fréjus | 22 June 2021 | 43.418976 | 6.74892 |
| PMR VP5852 | *Z. zebrus* | 3 | Port-Hercule, Zone E, Monaco | 2 July 2021 | 43.732882 | 7.424094 |
| PMR VP5853 | *Z. pallaoroi* | 1 | Port Gardian, Saintes-Maries-de-la-mer | 12 June 2018 | 43.449684 | 4.422563 |
| PMR VP5854 | *Z. pallaoroi* | 1 | Plage des Aresquiers, Frontignan | 14 May 2022 | 43.453062 | 3.815398 |
| PMR VP5855 | *Z. pallaoroi* | 9 | Marseillan Plage D, Marseillan | 27 June 2022 | 43.318023 | 3.557511 |
| PMR VP5856 | *Z. pallaoroi* | 6 | Camille Rayon Golf Juan C, Vallauris (Golfe-Juan) | 22 June 2022 | 43.567066 | 7.079608 |
| PMR VP5857 | *M. macrocephalus* | 5 | Camille Rayon Golf Juan C, Vallauris (Golfe-Juan) | 22 June 2022 | 43.567066 | 7.079608 |
| PMR VP5858 | *Z. zebrus* | 1 | Camille Rayon Golf Juan C, Vallauris (Golfe-Juan) | 22 June 2022 | 43.567066 | 7.079608 |
| PMR VP5859 | *M. macrocephalus* | 2 | Port-Hercule, Zone E, Monaco | 15 June 2022 | 43.732882 | 7.424094 |
| PMR VP5860 | *Z. pallaoroi* | 2 | Marseillan Plage E, Marseillan | 27 June 2022 | 43.318501 | 3.557346 |
| PMR VP5862 | *Z. pallaoroi* | 1 | Canet A, Canet-en-Roussillon | 10 June 2022 | 42.702469 | 3.039138 |
| PMR VP5863 | *M. macrocephalus* | 2 | COGE, Cogolin | 6 June 2022 | 43.264854 | 6.589862 |
| PMR VP5864 | *M. macrocephalus* | 4 | Fréjus A, Fréjus | 8 June 2022 | 43.418976 | 6.74892 |
| PMR VP5865 | *Z. pallaoroi* | 2 | Fréjus A, Fréjus | 8 June 2022 | 43.418976 | 6.74892 |
| PMR VP5866 | *Z. pallaoroi* | 3 | LONDE A, La-Londe-les-Maures | 21 June 2022 | 43.114968 | 6.247295 |
| PMR VP5867 | *Z. pallaoroi* | 5 | Camille Rayon Golf Juan A, Vallauris (Golfe-Juan) | 22 June 2022 | 43.562951 | 7.077255 |
| PMR VP5869 | *Z. pallaoroi* | 1 | Marseillan Plage C, Marseillan | 27 June 2022 | 43.317492 | 3.55792 |
| PMR VP5870 | *Z. pallaoroi* | 2 | ISSAM A, Roquebrune-sur-Argens (Les Issambres) | 7 June 2022 | 43.339601 | 6.685527 |
| PMR VP5871 | *Z. zebrus* | 3 | ISSAM A, Roquebrune-sur-Argens (Les Issambres) | 7 June 2022 | 43.339601 | 6.685527 |
| PMR VP5872 | *Z. pallaoroi* | 3 | Fontvieille A, Monaco | 15 June 2022 | 43.729357 | 7.422577 |

**Table 2.** Morphometric characters of *Zebrus pallaoroi* and *Zebrus zebrus* used in the diagnosis of *Zebrus pallaoroi*. Figures are rounded to two decimal places. SL + CL: standard length + caudal-fin length; SN/E: eye diameter in snout length; PN/AN: anterior nostril length in posterior nostril length; HL/E: eye diameter in head length; AM/VI: length of the ventral spine in anterior membrane depth at the midline; $\rho\theta/\alpha$: diameter of the head canal pore $\alpha$ in the distance between pore $\rho$ and $\theta$; SL/VD: body depth at the pelvic-fin origin in standard length. The ranges of values presented in the species description [6] are given for comparison.

| Species | *Zebrus pallaoroi* | | | *Zebrus zebrus* | | |
|---|---|---|---|---|---|---|
| **Sex** | **Males** | **Females** | **[6]** | **Males** | **Females** | **[6]** |
| **Number of Specimens** | **33** | **32** | | **6** | **4** | |
| SL + CL (mm) | 20.97, CL damaged to 44.06 + 11.41 | 22.07 + 5.06 to 34.19 + 8.13 | | 21.31 + 5.55 to 30.17 + 7.19 | 22.01 + 5.96 to 25.36 + 5.76 | |
| SN/E | 0.98–1.33 | 0.98–1.49 | 1.1–1.2 | 0.73–0.94 | 0.71–0.99 | 0.8–0.9 |
| PN/AN | 0.56–1.00 | 0.60–1.00 | 0.7–0.9 | 0.21–0.48 | 0.25–0.50 | 0.25–0.5 |
| HL/E | 4.14–5.65 | 4.17–5.36 | 4.2–4.7 | 3.41–4.11 | 3.41–4.47 | 3.1–4.1 |
| AM/VI | 0.53–0.78 | 0.50–0.77 | 0.61–0.70 | 0.26–0.41 | 0.19–0.36 | 0.26–0.50 |
| $\rho\theta/\alpha$ | 2.00–4.32 | 2.56–4.74 | about 3 | 1.68–3.50 | 1.90–2.59 | about 2 |
| SL/VD | 4.34–5.40 | 4.27–5.69 | 5.3–5.4 | 4.03–4.79 | 3.69–4.39 | 4.0–4.7 |

In each of the three species, males showed a greater variation in size, with both the smallest and also the largest recorded individuals being males. The largest males were 28.9%, 19.0%, and 16.3% larger than the largest females for *Z. pallaoroi*, *Z. zebrus*, and *M. macrocephalus*, respectively (Table 2). We found a limited but significant sexual size dimorphism for *Z. pallaoroi* (Welch's *t*-test: t = 4.29; $p < 10^{-4}$; we used Welch's rather than Student's test because of unequal variance between males and females as revealed by Levene's test of variance homogeneity), and for *M. macrocephalus* (Welch's *t*-test: t = 2.64; $p = 0.017$). However, we did not find any significant sexual size dimorphism for *Z. zebrus* (Welch's *t*-test: t = 1.49; $p = 0.177$), but the low sample size (six males and four females) renders this result unreliable.

### 3.2. Geography

*Zebrus pallaoroi* was found in 19 out of the 22 samples, and in 10 out of 12 localities (Figure 1, Table 1). Of these 10 localities, 8 were part of the maintenance of Biohut© habitats, and the other 2 involved specimens were collected in their natural habitat (see Sampling in Section 2). The other two species were less frequent: *Z. zebrus* was found in 5 samples and *M. macrocephalus* in 6 samples, out of the 20 Biohut© samples (Table 1).

### 3.3. Ecology

The two deepest specimens were collected in their natural habitat and were found at 3 m of depth at Plage du Prévost, Palavas-les-Flots (PMR VP5841) and at 5 m of depth at Plage des Aresquiers, Frontignan (PMR VP5854) (Figure 3, Table 1). In addition, PMR VP5841 was found under a solitary rock in a vast expanse of sand, and PMR VP5854 under a stone in a field of pebbles surrounded by sand. The Biohut© revealed the species' affinity for small artificial habitats made of steel cages filled with oyster shells and placed on the bottom or hung from the man-made construction at depths 0.5 and 1 m. Another striking finding with the Biohut© is that, of the 16 samples including *Z. pallaoroi*, 1 also included *Z. zebrus* only, 2 also included *M. macrocephalus* only, and 2 included all three species together (PMR VP5849, PMR VP5850, and PMR VP5851 at Fréjus A on 22 June 2021, and PMR VP5856, PMR VP5857, and PMR VP5858 at Camille Rayon Golf Juan C on 22 June 2022) (Table 1).

*3.4. Morphology*

The studied characters of *Z. pallaoroi* revealed substantial new findings between the present study and the original description. Overall, the number of character states was larger, and the character ranges were wider for both species in the present data compared to in the original description and showed a larger overlap between the two *Zebrus* species (Table 2, Figure 4). Reassuringly, however, the original diagnosis remained useful when considering the characters altogether. Indeed, the identification of *Z. pallaoroi* and *Z. zebrus* was never in doubt, as for each individual, we observed a clear majority of characters whose value (state or quantitative measure) is diagnostic of either *Zebrus pallaoroi* or *Z. zebrus*. In more detail, we found that only a few characters in individuals we assigned to *Zebrus pallaoroi* had values compatible with *Z. zebrus* (as presented in the original description, [6]): 17 individuals 0/11, 22 individuals 1/11, 23 individuals 2/11, 3 individuals 3/11, 0 individuals >3/11. Similarly, for individuals we assigned to *Zebrus zebrus*, the number of characters whose value was compatible with *Z. pallaoroi* was distributed as follows: 3 individuals 0/11, 7 individuals 1/11, 0 individuals >1/11. Seventy-four of the 77 character values observed in *Z. pallaoroi* individuals, which were compatible with *Z. zebrus* according to the original description, belonged to these three characters: body depth at the pelvic-fin origin in standard length (SL/VD; 16 *Z. pallaoroi* individuals showed values compatible with *Z. zebrus*; graph 11 in Figure 4); count of all lateral bands (20 *Z. pallaoroi* individuals showed values compatible with *Z. zebrus*, graph 9 in Figure 4); and most notably, the width of the lateral dark bands at the lateral midline compared to the width of the pale interspaces at the lateral midline (38 *Z. pallaoroi* individuals showed values compatible with *Z. zebrus*, graph 8 in Figure 4). The ability of the first and second characters to discriminate *Z. pallaoroi* had already been critiqued in [6,16], respectively. When excluding these three characters, the number of *Z. pallaoroi* individuals with *Z. zebrus* character values is reduced to only three individuals, each with a single contradictory character out of eight characters. For all these reasons, we consider our morphological identifications highly reliable.

The number of strictly non-overlapping characters decreased from eight in the original description to three in the present study. Those three characters are "anterior nostril length in posterior nostril length (PN/AN)"; "the presence of a short anterior transverse ridge that connects left and right ventrolateral head ridges"; and "the length of the ventral spine in anterior membrane depth at the midline." Three additional characters showed very limited overlap. The overlap involved extreme values only and was based on a single individual for each character. These three characters, which can thus be considered rather reliable for identifying a large majority of *Z. pallaoroi* individuals, are as follows: "eye diameter in snout length", "the length of the suborbital sensory papillae row *c5i* compared to the distance between row *c5i* and row *d*", and "the count of lateral dark bands in front of the vertical of the second dorsal fin". Three characters remain useful for confirming *Z. pallaoroi* in most individuals: "eye diameter in head length" (graph 3 in Figure 4), "the diameter of head canal pore $\alpha$ in the distance between pore $\rho$ and $\theta$ ($\rho\theta/\alpha$)" (if one *Z. zebrus* outlier is excluded from the *Z. zebrus* range, graph 6 in Figure 4), and "body depth at the pelvic-fin origin in standard length" (graph 11 in Figure 4). The last character was deemed unreliable by [6], and was not included in the species diagnosis, but we consider it of potential interest in this study. Furthermore, in these three characters, overlapping values did not correlate within individuals. In other words, there was no tendency for an individual with one overlapping character value to also have the other two values within the overlapping range. To more quantitatively test whether the overlap in the three variables tends to co-occur within individuals, we created variables describing, for each individual, whether the value is overlapping or not (binary variable) and applied chi-square tests of independence between the variables. We did not find a significant association between the overlap statuses for any of the three pairs of variablesF (HL/E vs. $\rho\theta/\alpha$: $p = 0.774$; SL/VD vs. $\rho\theta/\alpha$: $p = 0.112$; HL/E vs. SL/VD: $p = 0.793$).

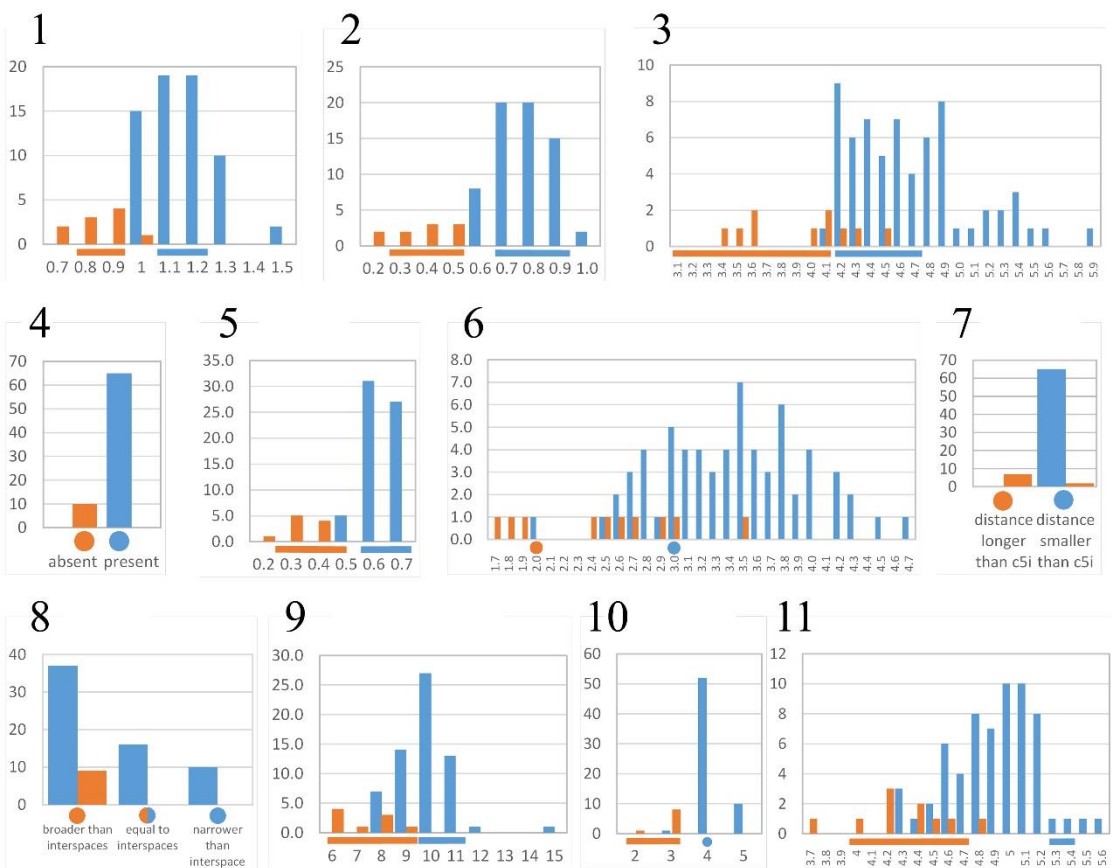

**Figure 4.** Diagnostic character values recorded in 65 collected individuals of *Zebrus pallaoroi* (▮) and 10 collected individuals of *Zebrus zebrus* (▮): (**1**) eye diameter in snout length (SN/E); (**2**) anterior nostril length in posterior nostril length (PN/AN); (**3**) eye diameter in head length (HL/E); (**4**) short transversal ridge connecting the left and right ventrolateral head ridges transversally on the anterior part present or absent; (**5**) length of the ventral spine in anterior membrane depth at the midline (AM/VI); (**6**) head canal pore α diameter (α) and distance between pores ρ and θ (ρθ); (**7**) length of the suborbital sensory papillae row *c5i* compared to the distance between row *c5i* and row *d*; (**8**) lateral dark band widths compared to pale interspace widths; (**9**) total count of lateral dark bands on the body; (**10**) count of lateral dark bands in front of the vertical of the second dorsal fin; (**11**) body depth at the pelvic-fin origin in standard length (SL/VD). The size classes of the morphometric characters are based on the data rounded to one decimal place. The published values from the species description for *Zebrus pallaoroi* (━, •) and *Zebrus zebrus* (━, •) are presented below the graphs as horizontal bars for ranges or as dots for single values [6].

The final two characters showed substantial overlap between both *Zebrus* species. The first is "the total count of lateral dark bands on the body counted at the lateral midline", which had been previously recognized as strongly overlapping [16] (graph 9 in Figure 4). The second is "the width of the lateral dark bands at the lateral midline compared to the width of the pale interspaces at the lateral midline," which showed similar character values for both *Zebrus* species in the present sample and thus has no differential use (graph 8 in Figure 4).

Using chi-square tests on both *Zebrus* species considered together, we did not find any sexual difference in the number of overlapping values, for any of the quantitative variables (SN/E: $p = 0.874$; PN/AN: $p = 1$; HL/E: $p = 0.221$; AM/VI: $p = 1$; ρθ/α: $p= 0.537$; count of dark lateral bands on the body along the lateral midline: $p = 0.220$; count of dark lateral bands in front of the vertical of the second dorsal fin: $p = 0.898$, SL/VD: $p = 0.613$). Additionally, for these characters, the number of overlapping values was not correlated with the standard length (point-biserial correlation between the continuous variable SL

and binary variables of overlap: SN/E: r = −0.195, $p = 0.093$; PN/AN: no variance; HL/E: r = −0.230, $p = 0.059$; AM/VI: no variance; $\rho\theta/\alpha$: r = 0.188, $p = 0.107$; count of dark lateral bands on the body along the lateral midline: r = −0.167, $p = 0.153$; SL/VD: r = −0.195, $p = 0.093$). One exception is the number of dark lateral bands in front of the vertical of the second dorsal fin, for which the likelihood of having an overlapping value decreases as SL increases (r = −0.281, $p = 0.015$).

*3.5. Molecular Analysis*

A BLAST of the cytochrome b sequence from individual PMR VP5841 against the NCBI nucleotide library returned four matches above 99% identity. These four individuals (Genbank accession n° from MW196294 to MW196297) were sampled in Greece and Montenegro by [6] and were all identified as *Zebrus pallaoroi*. The individual MW196297, notably, is the holotype of *Z. pallaoroi*. Thus, the mitochondrial lineage of PMR VP5841 can be unambiguously attributed to that species.

## 4. Discussion

In the present study, the identity of *Z. pallaoroi* was confirmed by morphology, and in one of two deeper records, it was also confirmed genetically. In the 65 individuals studied here, compared to the 3 individuals studied in the original description, the character ranges expanded considerably, with some overlap in the ranges between the two species (Table 2, Figure 4). Nevertheless, the original diagnosis remained resilient in accurately identifying the two *Zebrus* species, with *Z. pallaoroi* individuals predominantly showing the expected character states and ranges (see Section 3.4). Moreover, we found that qualitative characters can also fail to discriminate between the two species, just as quantitative ones can, for example, the categorial character describing whether the dark bands at the lateral midline are wider or narrower than the pale interspaces. It remains unclear whether the expansion of character ranges was due to the larger sample size or whether values were influenced by morphological differences between this remote population and those from the type localities. The small sample sizes analyzed in previous studies do not allow us to determine whether the distribution of character values has simply expanded (supporting the hypothesis of a sample size effect) or has shifted (suggesting the hypothesis of geographic variation) between the western Mediterranean and Adriatic samples. A larger sampling of *Zebrus* spp. in the Adriatic would help to better identify any potential geographic variation. In any case, the robustness of the original diagnosis of *Z. pallaoroi* ([6,17] and present data) has provided insights for future species descriptions based on small sample sizes. (1) Although the intrageneric diagnosis was a differential diagnosis against only one other congeneric species, Kovačić et al. [6] used eight characters, far exceeding the recommendation of a single distinguishing character against each species [3]. (2) At the time of description, all characters were non-overlapping between *Z. pallaoroi* and *Z. zebrus* [6]. (3) Non-conventional characters in gobiid descriptions, such as the proportions between parts of the head lateral line system, were found to be distinctive and were included in the diagnosis [6]. (4) Previously and commonly used characters in gobiid descriptions were measured with greater resolution, such as transitioning from semi-quantitative to quantitative measurements in the ratio between the nostrils or in the anterior membrane and pelvic-spine ratio [6]. (5) Characters previously used in gobiid descriptions were applied using precisely defined measuring positions, consistent with the methods used in [6], and these positions are clearly expressed and illustrated in the present study.

It is inevitable that new gobiid species will continue to be described based on a limited number of individuals. Waiting for additional samples could delay the description of new species for decades in some cases, which is also undesirable. For example, the Mediterranean goby *Didogobius bentuvii* Miller, 1966, is still known only from the holotype collected in 1962 [13]. In hindsight, the decision to describe the species based solely on the holotype was justified. For species described in this way, there is a risk that the proposed

differential diagnosis may become obsolete as more individuals are examined, potentially requiring a species redescription. However, we believe these limitations are outweighed by the benefits of describing new species, particularly those that are rare or cryptic and may have urgent conservation needs. Nevertheless, some recommendations could be followed to improve the likelihood that a differential diagnosis will continue to effectively distinguish the species from similar or closely related species of Gobiidae, even as additional populations are studied [4]. Therefore, our recommendations for ensuring that diagnostic characters based on small samples remain functional and resilient as the number of studied individuals increases are as follows. (1) Use several characters and not just one, both qualitative and quantitative, to distinguish the species from each of its closest relatives, as some overlap in character values with sister species is likely to invalidate some of them as sample sizes increase. (2) Focus on only non-overlapping characters, if possible, for the same reason. (3) Search for non-conventional characters that have not previously been used in descriptions of congeneric or related species. (4) Quantify characters or increase the resolution of already quantitative measurements. (5) Precisely define measurement positions for the quantitative characters being studied.

In addition to serving as evidence at the time of species description, the subsequent role of diagnoses as the most convenient among tools for fish identification (see the Introduction) should also be considered. Thus, overlapping characters may still be useful for identifying certain parts of a population in future studies, but the original diagnoses based on small samples should avoid relying on them. Unfortunately, nature is rarely aligned with our needs, and cases exist where valid and justified fish species lack clear morphological differential characters. Numerous cases of overlapping diagnostic characters between congeneric species have been documented in European freshwater fishes [3]. In Mediterranean marine fishes, diagnostic characters also overlap with closely related species in some species descriptions [21], sometimes to the extent that morphological differentiation between species is nearly impossible for every individual, despite genetic and geographical distinctions [22]. The absence of completely discriminative characters in diagnoses combined with a low number of overlapping or highly overlapping characters makes species identification based on such diagnoses unreliable. A taxonomist can only do their best on a case-by-case basis, without any guarantee that a functional diagnosis in new species descriptions will be established every time. Statistically significant differences in character state frequencies or value distributions are of little use for taxon identification, as they may exist even in heavily overlapping frequencies and distributions. When used alone without genetic data in species descriptions, that kind of evidence is also questionable for recognizing species, as significant morphological differences can also occur between distinct populations of the same species or due to non-representative sampling. However, in some cases, statistically significant differences are the only morphological distinctions that can be extracted and presented between species found to be distinct from genetics [22]. This situation should be clearly stated in the species description, i.e., that the species is established on genetic evidence alone, and anything useful for morphological distinction should be also provided. The lack of studies examining the influence of small sample sizes on the quality and longevity of species diagnoses is not surprising, given that zoological taxonomy still grapples with more fundamental issues related to diagnoses. These include the complete absence of a diagnosis in some species descriptions, or the provision of a non-differential diagnosis that is simply a shortened version of the description without adequate comparison to similar species, resulting in diagnoses that fail to be contrastive and/or state-specific [4,7]. Furthermore, the misuse of diagnoses is one aspect of the broader taxonomical malpractices that persist, or are even increasing, as discussed in [4].

In this study, the maximum known size of a male *Z. pallaoroi* was increased to over 55.5 mm in total length, compared to the total length of 47.7 mm known until the present research [6] and similar to the maximum known length of the other two species. The known maximum size of *Z. zebrus* is 61.0 mm [23], and 51.5 mm for *M. macrocephalus* [11] (Table 2). However, for *Z. zebrus*, the known total length of positively identified individuals after



the description of *Z. pallaoroi* is considerably smaller, 34.9 mm in [6] and 37.4 mm in the present research. Overall, the identification of *Z. zebrus* in studies conducted before 2021 should be approached with caution, considering the possibility of misidentified *Z. pallaoroi* individuals, including those larger than the maximum size recorded for *Z. zebrus* after 2021.

Genetic and morphological analyses revealed the presence of *Z. pallaoroi* in southern France and Monaco. These records extend the known geographic range of this species from the eastern to the western Mediterranean. The region near the French–Spanish border now represents the westernmost area with confirmed records of *Z. pallaoroi*. In comparison, *Z. zebrus* and *M. macrocephalus* have a broader known geographic distribution. *Z. zebrus* has been confirmed from the Atlantic coastline of southwest Spain in the west to the eastern Aegean Sea and the Levantine Sea in the eastern Mediterranean, as well as in the Bosphorus Strait and the Black Sea in the east (reviewed in [11]). However, several unidentified *Zebrus* records have been reported from the Atlantic coast of Portugal and France (photos available on iNaturalist for Portugal, pers. com. with photos to J.R. for France), as well as in Turkey and along the African Mediterranean coast, particularly in Tunisia (J.R., pers. obs. with photos). These findings suggest that the known geographic distribution of these two species is likely to change as more individuals are collected and examined. Lastly, *M. macrocephalus* ranges from Mar Menor in Spain in the west to the eastern Aegean Sea and the Levantine Sea in the eastern Mediterranean, and in the Black Sea and southern Mediterranean, excluding the Levantine Sea (reviewed in [11]).

One of the most notable findings of this study is that *Z. pallaoroi*, *Z. zebrus*, and *Millerigobius macrocephalus* were found cohabiting within the small, homogeneous interstices of the same artificial habitats (Biohut©). This was relatively unexpected, given that all three species are phylogenetically closely related and so similar in appearance that they can barely be distinguished without the use of stereomicroscopic magnification (though this is less true for *Millerigobius*, which can now be convincingly identified with good-quality in situ photographs [6,11]). This syntopy raises questions about the relevance of the competitive exclusion principle for these taxa [24]. Nonetheless, we cannot rule out the possibility of some form of micro-habitat or niche specialization. The artificial Biohut© system represents an excellent opportunity to investigate this question in future research.

Two *Z. pallaoroi* specimens were found in habitats mixed with soft sediment: PMR VP5841 was found under a solitary rock in a vast sandy area, and PMR VP5854 was found under a stone in a pebble field surrounded by sand. These findings differ from the species' known habitat, which includes mediolittoral gravel, cobbles, boulders, or bedrock. Outside the habitats examined in this study, *Z. zebrus* and *M. macrocephalus* exhibit broader habitat preferences compared to *Z. pallaoroi*. In addition to the fields of gravel, cobbles, boulders, bedrock, mixed rocky and sandy bottoms, and man-made structures where each of three species has been recorded, *Z. zebrus* also occurs in *Cymodocea* or *Posidonia* seagrass meadows, coastal lagoons, and under sea urchins for juveniles (reviewed in [11]), while *Millerigobius* also inhabits transitional waters, either hypersaline or brackish (reviewed in [11]).

*Z. pallaoroi*, along with its two sister species to a certain extent, displayed a surprising abundance and frequency of occurrence in the small artificial habitats made of steel cages filled with oyster shells. The two specimens of *Z. pallaoroi* collected in their natural habitat were found at previously unrecorded depths: PMR VP5841 was found at a depth of 3 m at Plage du Prévost, Palavas-les-Flots, and PMR VP5854 at a depth of 5 m at Plage des Aresquiers, Frontignan, expanding the known depth range for this species from "mediolittoral" to "mediolittoral and upper infralittoral" (Figure 3, Table 1). These deeper records of *Z. pallaoroi* prompt us to reconsider its status as a stenobathic shallow water specialist, like the Mediterranean clingfish *Gouania* spp., and to reassess the use of depth as a distinguishing feature between this species and *Z. zebrus* and *M. macrocephalus*. However, *Z. zebrus* and *M. macrocephalus* still exhibit a wider known depth range than *Z. pallaoroi*: 0.1 to 36 m for *Z. zebrus* and 1 to 25 m for *M. macrocephalus* (reviewed in [11]). The present samples of *M. macrocephalus*, collected at 0.5 m, represent the shallowest published record of the species.

Based on the present results and the suggestions outlined above, we propose the following revised diagnosis of *Z. pallaoroi*. *Z. pallaoroi* differs from its only congeneric species, *Z. zebrus*, in each of the following character values: (1) snout longer or rarely equal to the eye, with a length 0.98–1.48 times the eye diameter, where values above 1 show no overlap with *Z. zebrus*; (2) posterior nostril in the form of a short tube, 0.56–1.00 of the anterior nostril; (3) eyes moderately small, eye diameter in head length of 4.14–5.65, where values above 4.5 show no overlap with *Z. zebrus*; (4) left and right ventrolateral head ridges connected transversally by a short ridge on the anterior part; (5) depth of the pelvic anterior membrane at the midline of 0.50–0.77 of the depth of the pelvic spinous ray; (6) head canal pores large, diameter of pore $\alpha$ of 2.00–4.74 in the distance between pore $\rho$ and $\theta$, where values above 3.5 show no overlap with *Z. zebrus*; (7) suborbital sensory papillae row *c5i* extending downwards to or near the level of row *d*, with the distance between row *c5i* and row *d* either absent or much smaller than the length of row *c5i* (a single *Z. zebrus* outlier was recorded with this character state); (8) body depth at the pelvic-fin origin of 4.24–5.69 in standard length, where values above 4.8 show no overlap with *Z. zebrus*; and (9) 4–5 lateral dark bands in front of the vertical of the second dorsal fin (a single *Z. pallaoroi* outlier was recorded with three bands).

**Author Contributions:** Conceptualization, M.K. and J.P.R.; methodology, M.K., L.B., and J.P.R.; software, J.P.R.; validation, M.K. and J.P.R.; formal analysis, M.K. and J.P.R.; investigation, M.K., L.B., and J.P.R.; resources, M.K., L.B., and J.P.R.; data curation, M.K. and J.P.R.; writing—original draft preparation, M.K.; writing—review and editing, M.K., L.B., and J.P.R.; visualization, M.K. and J.P.R.; supervision, J.P.R.; project administration, M.K.; funding acquisition, M.K., L.B., and J.P.R. All authors have read and agreed to the published version of the manuscript.

**Funding:** M.K. was funded by the grant of the Croatian Science Foundation under the project IP-2022-10-7542.

**Institutional Review Board Statement:** The sampling scheme followed a standardized protocol approved by international authorities (EU/DG Mare, FAO/GFCM). If a live specimen of a species subject to conservation measures was caught, it was quickly sampled (4–5 min) and returned back to the sea unharmed, giving it a chance for survival, following the recommendation GFCM/36/2012/3 (http://www.gfcmonline.org/decisions/, accessed on 1 October 2024) on fisheries management measures for the conservation of sharks and rays in the GFCM area. Cryptobenthic fish were euthanized by administering an overdose of anaesthetic in compliance with the recommendation of Decree Law n. 26 of 4 March 2014. All efforts were made to minimize suffering.

**Informed Consent Statement:** Not applicable.

**Data Availability Statement:** The data presented in this study are available on request from the corresponding author.

**Acknowledgments:** We thank Laure Benoit for her help with genetic analyses.

**Conflicts of Interest:** The authors declare no conflicts of interest.

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
