# Peer review of "Testing of the Taxonomic Diagnosis of Zebrus pallaoroi Kovačić, Šanda & Vukić, 2021 (Actinopteri: Gobiiformes: Gobiidae), on a Large Sample from the Western Mediterranean"

_fishes, doi:10.3390/fishes9120511_

Round 1
Reviewer 1 Report
Comments and Suggestions for Authors
Dear authors,
Congratulations for your work results.
Please find below some suggestions which from my point of view can improve some aspects of the paper:
0. The title is confusing in its goal. In my opinion the title is over the results which this study can offer. In another words is not possible to really on one single species to offer advices for a whole taxonomic group, the fish or even more. The title must be readjusted. To use this in the title you need an exhaustive review not species study case only.
1. The citations and references can be enriched.
2. The not original texts of the paper should have citations and references.
3. Please highlight the limitations of your approach and methods.
4. Unclear here: We used 11 morphological characters to identify Z. pallaoroi. These included eight characters from [12], with the lateral dark brown bands on the body decomposed in two separate characters
5. Please explain why you chose the studied characters and not others or many others.
6. The eleven studied characters of Z. pallaoroi revealed substantial differences between the present study and the original description. Rephrasing? ... revealed substantial addings/new findings not differences may be?
7. In the 65 individuals studied here, compared to the three individuals studied in the original description, the character ranges expanded considerably, with some overlap in the ranges between the two species. I think this is normal when you compare such a different pool of individuals number. The question is if 65 is still enough. If you will repeat the study on much many of them you can also improve the results? Where is the optimum limit of individuals number and why?
8. So true:) A taxonomist can only do their best on a case‐by‐case basis, without any guarantee that every time the functional diagnosis in new species descriptions will be established.
9. Laure Benoit did not deserve to be co-author? Acknowledgments: We thank Laure Benoit for genetic analyses.
10. May be if the paper focus on the studied species and only on Gobiidae group, it will have more confusing and redundant info, and more more clear specific value of the results.
All the best
Reviewer
Author Response
Author's Reply to the Review Report (Reviewer 1)
Comment 1: The title is confusing in its goal. In my opinion the title is over the results which this study can offer. In another words is not possible to really on one single species to offer advices for a whole taxonomic group, the fish or even more. The title must be readjusted. To use this in the title you need an exhaustive review not species study case only.
Response 1: We have modified the title in the revised version.
Comment 2: The citations and references can be enriched.
Response 2. The following references related to the topic have been added in the Introduction and in the Material and Methods:
Balakrishnan R. Species concepts, species boundaries and species identification: a view from the tropics. Systematic Biology 2005;54:689–93. https://doi.org/10.1080/10635150590950308
Borkent A. Diagnosing diagnoses—can we improve our tax onomy? Zookeys 2021;1071:43–8. https://doi.org/10.3897/ zookeys.1071.72904
Dubois A (2017) Diagnoses in zoological taxonomy and nomenclature. Bionomina 12: 63–85. https://doi.org/10.11646/bionomina.12.1.8
Renner, S.S. A return to Linnaeus’s focus on diagnosis, not description: the use of DNA characters in the formal naming of species. Systematic Biology, 2016, 65, 1085–1095. https://doi.org/10.1093/sysbio/syw032 PMID:27146045
Comment 3: The not original texts of the paper should have citations and references.
Response 3: Citations have been added throughout the Introduction so that every non original claim is supported by a reference (“However, species descriptions should also include a diagnosis [2]. “; “The differential diagnosis is a formal statement of the characters that distinguish a given taxon from other specifically mentioned taxa [2]. The differential diagnosis is part of diagnosis [2].”; “The differential characters against particular taxa can also be presented in details separately after diagnosis or even in the “Remarks” section of a species description, with the comparison of character values and states of new taxa against other specifically mentioned taxa, e.g. [6].”; “In the current practice in gobiology, a new species is established as valid mostly based on morphological (including coloration) and molecular evidences that distinguish it from closely related species (the examples of gobiid species description reviewed in [10]).“; „Out of the 13 Mediterranean gobiid species described in the 21st century, only the earliest one has a description including just morphological and coloration characters as evidence (reviewed in [10] and [11]).”; “However, the prevailing practice when describing new European marine gobiid species is to list differential morphological and coloration characters in the species diagnosis, while molecular evidence are presented in a separate part of the species description (reviewed in [10] and [11]).”; “The morphological diagnosis usually contains the most useful information for later identification of new and additional material [5].”; “Indeed, while the diagnosis is more complete and thus more reliable than identification keys [5], it is significantly shorter, less complex, faster and easier to use than the complete morphological description or the molecular methods [4,5].”; “Ideally, diagnostic characters should have states (for qualitative characters) and ranges (for quantitative characters) that do not overlap between the compared species [4].”; “For example, in all 13 gobiid species described since the beginning for the 21st century for the Mediterranean Sea, the external morphology was described from the type material only (reviewed in [10] and [11]).”).
Comment 4: Please highlight the limitations of your approach and methods.
Response 4: Limitations have now been highlighted when presenting the aims (“The objectives of this paper are to report the first record of Z. pallaoroi in the Western Mediterranean, to expand knowledge of the species’ geographical and ecological characteristics, and to test and revise its species diagnosis, originally based on the limited number of only diagnostic morphological characters, using a large sample from a population geographically distant from the type locality.”).
Comment 5: Unclear here: We used 11 morphological characters to identify Z. pallaoroi. These included eight characters from [12], with the lateral dark brown bands on the body decomposed in two separate characters
Response 5: This has been clarified in the revised manuscrit (“We used 11 morphological characters actually used or just considered for Z. pallaoroi morphological diagnosis [6,17] to identify Z. pallaoroi . These included eight characters from [6], though one of these characters, “the lateral dark brown bands on the body”, was split into two separate characters: the number of dark lateral bands (photo 9 in Figure 2) and the mean width of the dark lateral bands compared to the mean width of the pale interspaces (photo 10 in Figure 2), resulting in nine characters. The tenth character is body depth at pelvic-fin origin, mentioned as non-overlapping between Zebrus species in the studied material in the original species description of Z. pallaoroi [6], but overlapping according to Miller’s [15] values for Z. zebrus (photo 11 in Figure 2) and therefore not use in [6]. Finally, the eleventh character was the count of dark lateral bands in front of the second dorsal fin, which was proposed in [17] to replace the total count of dark lateral bands originally proposed in [6] (photo 8 in Figure 2).”).
Comment 6: Please explain why you chose the studied characters and not others or many others.
Response 6: Testing the species diagnosis was our sole and primary goal, as stated in the Title and Abstract (“The diagnostic characters of Z. pallaoroi, originally based on a limited number of type specimens, were tested on this larger sample and critically analyzed.”), and reiterated throughout the text of the manuscript. Therefore, we used only the original characters from the species diagnosis in the species description, along with the characters suggested useful for diagnosis in subsequent work.
Comment 7: The eleven studied characters of Z. pallaoroi revealed substantial differences between the present study and the original description. Rephrasing? ... revealed substantial addings/new findings not differences may be?
Response 7: This sentence has been removed from the revised manuscript.
Comment 8: In the 65 individuals studied here, compared to the three individuals studied in the original description, the character ranges expanded considerably, with some overlap in the ranges between the two species. I think this is normal when you compare such a different pool of individuals number. The question is if 65 is still enough. If you will repeat the study on much many of them you can also improve the results? Where is the optimum limit of individuals number and why?
Response 8: The character range increases here could be due to the sample size, but also due to the statistical representativeness of the sample for all populations of the species. Regarding statistical representativeness, we view regional differences among populations as the possible explanation for why the examined material differs from previously examined material. The current level of knowledge for this species, as for any other Gobiidae species, falls significantly short of what is needed to draw conclusions on the regional differences among populations. In the Discussion, the section stating “It remains unclear whether the expansion of character ranges was solely due to the larger sample size or if some values were also influenced by morphological differences between this remote populations and those from the type localities.”, is now further elaborated “The small sample sizes analyzed in previous studies do not allow us to determine whether the distribution of character values has simply expanded (supporting the hypothesis of a sample size effect) or has shifted (suggesting the hypothesis of geographic variation) between the Western Mediterranean and Adriatic samples. A larger sampling of Zebrus spp. in the Adriatic would help to better identify any potential geographic variation.”
Comment 9: So true:) A taxonomist can only do their best on a case‐by‐case basis, without any guarantee that every time the functional diagnosis in new species descriptions will be established.
Response 9: We thank the reviewer 1 for the comment and the shared understanding.
Comment 10: Laure Benoit did not deserve to be co-author? Acknowledgments: We thank Laure Benoit for genetic analyses.
Response 10: Laure Benoit only assisted one of the author (JPR) for the genetic analyses, in a proportion that traditionally does not entail co-authorship. This has been precised in the revised manuscript (“Acknowledgments: We thank Laure Benoit for her help with genetic analyses”)
Comment 11: May be if the paper focus on the studied species and only on Gobiidae group, it will have more confusing and redundant info, and more more clear specific value of the results.
Response 11: Several sentences have been modified in the revised manuscript, as follow: in the Abstract (“The diagnostic characters of Z. pallaoroi were revised, and recommendations were made for improving the study of diagnostic characters in gobiid species description, particularly when based on small sample sizes.”), Key contribution (“The recommendations were made for improving the study of diagnostic characters in gobiid fish species description, particularly when based on small sample sizes.”). Introduction (“In the current practice in gobiology, a new species is established as valid mostly based on morphological (including coloration) and molecular evidences that distinguish it from closely related species (the examples of gobiid species description reviewed in [10]).”; “However, the prevailing practice when describing new European marine gobiid species is to list differential morphological and coloration characters in the species diagnosis, while molecular evidence are presented in a separate part of the species description (reviewed in [10] and [11]).”; “Despite the potential weaknesses of gobiid species diagnoses based on small samples, to our knowledge, no published examples exist that re-evaluate or test these diagnoses once larger samples become available.”) and Discussion (“It is inevitable that new gobiid species will continue to be described based on a limited number of individuals”; “Nevertheless, some recommendations could be followed to improve the likelihood that a differential diagnosis will continue to effectively distinguish the species from similar or closely related species of Gobiidae, even as additional populations are studied [4].”) to restrict the subject, discussion and conclusion just to Gobiidae.

Reviewer 2 Report
Comments and Suggestions for Authors
Fishes 3310304
This paper reports a revision of diagnostic character states in a recently described goby species, based on a large sample size. The paper also suggests some guidelines for a reliable diagnosis, when a new species is going to be described. The paper is well written and useful for gobioid taxonomy. My only concern has to do with the unbalanced comparison between Zebrus pallaoroi and Zebrus zebrus, in terms of sample size: 65 individuals against 10 individuals. Further, the Z. zebrus samples seem to me smaller in terms of body size, as the higher classes were not present (perhaps due to the small sample size or to a bias in sampling or, alternatively, is Z. zebrus significantly smaller than the other species?). I’m not a taxonomist, so I can’t evaluate whether this aspect is a relevant problem in the context of this study, or not, but for sure umbalanced comparison is a statistical limitation. Once authors explain this aspect, the paper is suitable for publication in Fishes.
Author Response
Author's Reply to the Review Report (Reviewer 2)
Comment 1: This paper reports a revision of diagnostic character states in a recently described goby species, based on a large sample size. The paper also suggests some guidelines for a reliable diagnosis, when a new species is going to be described. The paper is well written and useful for gobioid taxonomy.
Response 1: We thank reviewer 2 for his comment and kind opinion.
Comment 2: My only concern has to do with the unbalanced comparison between Zebrus pallaoroi and Zebrus zebrus, in terms of sample size: 65 individuals against 10 individuals. I’m not a taxonomist, so I can’t evaluate whether this aspect is a relevant problem in the context of this study, or not, but for sure umbalanced comparison is a statistical
Response 2: The material was sampled, and after sampling each individual was identified as belonging to one of three similar species (M. macrocephalus, Z. pallaoroi, Z. zebrus). Therefore, it was not possible to balance the sample size, as the species composition could not be knonw in advance and controlled in this type of sampling.
The key outcome of the species identification was that the majority of identified individuals belonged to recently described Z. pallaoroi. This unbalanced composition allowed the current study on Z. pallaoroi, due to the large number of Z. pallaoroi individuals available for analysis.
The differences in sample size didn’t influence the outcomes of statistical analyses applied to sex-related or size-related changes, as no statistical comparison was made between the two species. Additionally, in the analyses conducted separately for each species the statistical methods were accounted for the sample size.
Finally, the ranges of morphological characters and the recorded character states from earlier small samples of Z. pallaoroi were compared with those obtained from the current larger sample of Z. pallaoroi (Table 2, Figure 4), and with the ranges in Z. zebrus. The Z. zebrus comparison included both the current sample of Z. zebrus individuals and a separate large sample of 25 Z. zebrus individuals studied in Kovačić et al. (2021). The inclusion of data from these 25 individuals mitigate the potential influence of the smaller sample size of Z. zebrus in the present study.
Comment 3: Further, the Z. zebrus samples seem to me smaller in terms of body size, as the higher classes were not present (perhaps due to the small sample size or to a bias in sampling or, alternatively, is Z. zebrus significantly smaller than the other species?).
Response 3: The sample of Z. zebrus in Kovačić et al. (2021) was moderately large, but the size of Z. zebrus individuals were still small. It is possible that only a very large sample, such as the one in Nieto & Alberto (1992), might include the larger size classes, assuming that individuals rarely survive to such sizes. However, there is also the possibility that large individuals previously recorded as Z. zebrus before the description of Z. pallaoroi in 2021 actually belonged to Z. pallaoroi, rather than Z. zebrus. This issue is addressed in the paragraph on fish size in the Discussion of the manuscript, which is now further elaborated (“Overall, the identification of Z. zebrus in studies conducted before 2021 should be approached with caution, considering the possibility of misidentified Z. pallaoroi individuals, including those larger than the maximum size recorded for Z. zebrus after 2021.”).
Reviewer 3 Report
Comments and Suggestions for Authors
In the present and interesting MS the authors reported on the presence of the recently described species Z. pallaoroi in the Western Mediterranean Sea, with precise description and analysis of morphological measures and traits distinguishing the species from other congenerics or other related taxa. The authors brought to the light the importance of specific diagnostic characters for Z. pallaoroi and highlighted the critical issue of having few samples.
Here are some very minor and major comments.
M&M
line 149: please, start the sentence with “Zebrus pallaoroi” and verify this style throughout the MS
line 152: please, find a synonim for “decomposed”.
My major concern is related to the paragraph starting at line 212: this section seems very reductive (I wonder if it is necessary at it is described). I see that deep analysis of species molecular taxonomy has been already explored in Kovačić et al. (2021), but I'd dedicate more care to the genetic approach (e.g. did the author check DNA extraction and PCR template on agarose gel? where the individual was sequences? Commercial provider/private lab/university? on which machine?).
Furthermore, there is no mention of genetic data analysis. Which software was used for the editing of the sequence? Then, I believe that 1 individual sequenced and BLASTed is reductive, but I also understand that molecular taxonomy is not the core of the MS. I suggest to complete the M&M section related to molecular applications with more detail and explain why did the author decided to analyse one individual (over 65!) first. In my mind, having the opportunity to explore the genetic variability of an entire pool of new individuals (far from the original basin where the new species was initially described) would be extremely interesting in terms of deepening the knowledge of their connectivity. I believe (and hope) that this will be the next-step.
Taking into consideration these elements I recommend the publication of this MS with minor revisions.
Comments on the Quality of English LanguageI found the quality of English Language good, clear and straightforward.
Author Response
Author's Reply to the Review Report (Reviewer 3)
Comment 1: In the present and interesting MS the authors reported on the presence of the recently described species Z. pallaoroi in the Western Mediterranean Sea, with precise description and analysis of morphological measures and traits distinguishing the species from other congenerics or other related taxa. The authors brought to the light the importance of specific diagnostic characters for Z. pallaoroi and highlighted the critical issue of having few samples.
Response 1: We thank reviewer 3 for his comment and kind opinion.
Comment 2: line 149: please, start the sentence with “Zebrus pallaoroi” and verify this style throughout the MS
Response 2: Corrected in the revised manuscript.
Comment 3: line 152: please, find a synonim for “decomposed”.
Response 3: The term “decomposed” has been replaced by “split” into the revised manuscript.
Comment 4: My major concern is related to the paragraph starting at line 212: this section seems very reductive (I wonder if it is necessary at it is described). I see that deep analysis of species molecular taxonomy has been already explored in Kovačić et al. (2021), but I'd dedicate more care to the genetic approach (e.g. did the author check DNA extraction and PCR template on agarose gel? where the individual was sequences? Commercial provider/private lab/university? on which machine?).
Furthermore, there is no mention of genetic data analysis. Which software was used for the editing of the sequence? Then, I believe that 1 individual sequenced and BLASTed is reductive, but I also understand that molecular taxonomy is not the core of the MS. I suggest to complete the M&M section related to molecular applications with more detail and explain why did the author decided to analyse one individual (over 65!) first. In my mind, having the opportunity to explore the genetic variability of an entire pool of new individuals (far from the original basin where the new species was initially described) would be extremely interesting in terms of deepening the knowledge of their connectivity. I believe (and hope) that this will be the next-step.
Response 4: We have added details on how the genetic analyses were conducted. Unfortunately, we do not currently have the budget to carry out the genetic analysis of all individuals. The sequencing of the single individual, which did not come from the pool of individuals collected in the Biohut, was conducted separately and opportunistically as part of another project conducted by a colleague. However, the tissues of all specimens have been preserved, and we hope to conduct the genetic study suggested by the reviewer in the near future.
Round 2
Reviewer 2 Report
Comments and Suggestions for Authors
Considering the responses of the authors, the paper is now acceptable for publication in Fishes.